# The Multifaceted Role of Connexins in Tumor Microenvironment Initiation and Maintenance

**DOI:** 10.3390/biology12020204

**Published:** 2023-01-28

**Authors:** Olga M. Kutova, Anton D. Pospelov, Irina V. Balalaeva

**Affiliations:** The Institute of Biology and Biomedicine, Lobachevsky State University of Nizhny Novgorod, 23 Gagarin Ave., Nizhny Novgorod 603950, Russia

**Keywords:** tumor microenvironment, connexins, cell–cell contacts, tumor stroma, carcinogenesis, tumor development, metastasis

## Abstract

**Simple Summary:**

Connexins are proteins which comprise gap junctions in cells. These junctions can directly connect neighboring cells and the cell interior with the extracellular microenvironment and thus they act as tissue integrators. Alterations in connexin regulation can lead to unfavorable shifts in the tissue adhesive context thus eradicating the constraints of the normal tissue microenvironment, triggering (or enhancing) cell motility. This review tries to examine the role of connexins in orchestrating the tumor microenvironment and hence their role in malignancy.

**Abstract:**

Today’s research on the processes of carcinogenesis and the vital activity of tumor tissues implies more attention be paid to constituents of the tumor microenvironment and their interactions. These interactions between cells in the tumor microenvironment can be mediated via different types of protein junctions. Connexins are one of the major contributors to intercellular communication. They form the gap junctions responsible for the transfer of ions, metabolites, peptides, miRNA, etc., between neighboring tumor cells as well as between tumor and stromal cells. Connexin hemichannels mediate purinergic signaling and bidirectional molecular transport with the extracellular environment. Additionally, connexins have been reported to localize in tumor-derived exosomes and facilitate the release of their cargo. A large body of evidence implies that the role of connexins in cancer is multifaceted. The pro- or anti-tumorigenic properties of connexins are determined by their abundance, localization, and functionality as well as their channel assembly and non-channel functions. In this review, we have summarized the data on the contribution of connexins to the formation of the tumor microenvironment and to cancer initiation and progression.

## 1. Introduction

Modern cancer research is focused on the identification of the mechanisms of cancer initiation, promotion and progression to further develop efficient treatment strategies. Accumulating experimental data testify to the tremendous role of the tissue microenvironment in tumor development and the formation of the unique tumor microenvironment (TME) [1,2,3]. TME is a complex and continuously evolving entity that includes immune cells, stromal cells, blood vessels, and non-cellular components such as extracellular matrix (ECM) and exosomes, and is characterized by specific physicochemical properties [4]. The crosstalk of TME constituents is executed by the simultaneous action of paracrine, autocrine, and endocrine signaling, and by the direct communication of adjacent and juxtaposed cells [5]. Direct intercellular communication and cell–matrix communication are maintained through intercellular junctions [6,7,8,9] and contacts with the extracellular matrix based on integrins and non-integrin receptors [10].

Connexins (Cxs) are proteins that form gap junctions (GJ) in cells of chordate organisms by assembling connexin monomers into hexameric hemichannels (connexins), that can function both autonomously, or undergo coupling to form a full-fledged channel that directly connects adjacent cells. To date 21 connexin isoforms have been discovered in humans [9]. The most widely used naming of connexin proteins is based on their predicted molecular weight in kilodaltons (e.g., Cx43, Cx32, etc.), whereas the classification of connexin genes was devised based on their sequence homology which falls into five subfamilies (α, β, γ, δ, ε) [11]. Connexins which have been found to play a significant role in cancer include Cx43 (the best studied), Cx25, Cx26, Cx32, Cx30, Cx31, Cx37, and Cx46.

Genes of connexins are shown to have one or multiple 5′UTR (untranslated region) exons which are separated from the exon containing the coding sequence by an intron of variable size. In some connexin genes, coding sequence may be interrupted by an additional intron. This allows a great variety of alternatively spliced connexin mRNA variants [12]. The transcription of connexin genes can be initiated by two promoters which are both regulated by universal and tissue-specific transcription factors. Among the universal regulators the most prominent are the Sp1, AP-1, cAMP and Wnt signaling pathways. Tissue specific regulation is provided by homeobox proteins, T-box and GATA transcription factors, hormones, etc. [13]. The production of connexins can be regulated epigenetically by hypermethylation of the promoter or acetylation/deacetylation of histones [14,15]. Concerning translation, connexins can be both silenced (via miRNA regulation) or facilitated (translation initiation via strong internal ribosome entry site (IRES)) [16,17].

The mutations in connexin gene family, both germline and somatic, have been established to be associated with multiple developmental abnormalities and syndromes. Nevertheless, the link between connexin mutations and cancer is still not clear. Several studies have revealed the association of certain type of tumors with mutated forms of connexins. Thus, the rate of mutated GJA1 gene (coding Cx43) is increased in colon adenocarcinomas [18], stomach adenocarcinomas and cutaneous melanoma [19]. Mutations of GJB2 (coding Cx26) might be associated with an increased propensity to develop skin cancer [20]; and germline mutations of this gene are assumed to increase the risk for early onset prostate cancer [21]. Mutated GJB6 (coding Cx30) and GJB7 (coding Cx25) might contribute to gastric and colorectal malignancies [22]. Cancer-specific rates of observed mutations support the important function of this group of proteins in the origin and development of tumors; however, their precise role and ability to drive carcinogenesis has to be further studied.

Connexins are tetraspan proteins, possessing four transmembrane domains, two extracellular loops and one intracellular loop; and intracellular N- and C-terminal tails (Figure 1). The transmembrane domains (TM1 to TM4) and amino terminus of connexins are relatively conserved, and form and maintain the connexin scaffold and transmembrane pore [23]. Early studies have shown that the vestibule of the connexin pore channels has a relatively large aperture ~40 Å, while it narrows in the membrane to ~15 Å due to a tilted domain orientation [24] and that the pore is formed by TM1 or TM3 [25,26,27]. Further studies have indicated that the N-terminus is also involved in the formation of the connexin funnel [28]. Recent cryo-EM studies examined the connexin structure with improved precision. In connexins formed by Cx46/50, the pore possesses a wide vestibule at the cytoplasmic side, formed by TM2 and TM3. The membrane-embedded part of the pore is maintained by TM1 and TM2 and lined with the N-terminal helix, responsible for channel gating. The extracellular part of the pore in the full-fledged gap junction is maintained by extracellular loop 1. The pore contains ordered water molecules both in solvent-accessible and buried sites which are thought to contribute to the regulation of channel specificity, conductance and gating [29]. Similar structural features are reported for full-fledged channels, based on Cx26 [30]. In the case of Cx31.3, its N-terminal helix does not line the pore, but resides in an entrance-covering position determine the pore’s very small diameter (8 Å against 10 and 11 Å in Cx26 and Cx46/50 channels, respectively). It should also be mentioned that Cx31.3 is not capable of forming full-fledged channels and its oligomerization is represented only by hemichannel formation [31]. Taking into account that the composition of the connexin scaffolds for Cx31.3 and Cx26 are nearly identical, this obligatory hemichannel state might be attributed to differences in the extracellular loops, affecting the backbone positioning [32].

The extracellular loops are the most conserved regions of the connexin molecule, and are responsible for connexins docking. They possess consensus patterns with three characteristic cysteines in each loop [33]. These cysteines are not susceptible to post-translational modifications, namely to S-nitrosylation, and there is an assumption that these cysteines may also act as extracellular redox sensors [34]. Additionally, point mutations artificially induced in the extracellular loops led to the failed coupling of Cx43 hemichannels [35] and a permanently closed state of Cx46 hemichannels [36], but with no affection on their synthesis, membrane trafficking or non-channel-related properties. The variability in connexins is mostly due to the cytoplasmic loop and C-terminal tail which act as a highly versatile platform for connexin regulation and connexin-mediated signaling. Regulation is performed by various post-translational modifications of the C-tail and signaling is realized through a wide range of interactions with the cytoskeleton and potential oncogenes (Figure 1B) [37]. The contribution of C-tail-mediated signaling in cancer progression is discussed further in the text.

Connexin oligomerization can occur by two routes, determined by a signature motif in a region of the cytoplasmic loop transition to TM3. The first route is characteristic of alpha-connexins: they are stabilized in a monomeric state by quality-control proteins and their further oligomerization takes place in the trans-Golgi network. This is determined by a conserved arginine or lysine residue in the cytoplasmic loop–TM3 transition region [38]. The second route is characteristic of beta-connexins. They tend to oligomerize rapidly due to a unstable monomeric state, as they possess a signature conserved di-tryptophan motif. Some connexins lack a signature motif (Cx37 and Cx40) [39,40]. Connexins can homo- and hetero-oligomerize. Additionally, the features of hetero-oligomerization are determined by the presence of two oligomerization pathways. Thus, alpha-connexins tend to hetero-oligomerize with each other and with connexins lacking the conserved motif; while beta-connexins tend to hetero-oligomerize exclusively with each other. This process is determined by the expression levels of proteins both in the case of alpha- and beta-connexins, as well as by the extent and timing of the stable monomeric state in the case of alpha-connexins [41]. The type of connexins which comprise the hemichannel or full-fledged channel and the composition (homo- or heteromeric) dictates pore size, permeability, and even the obligatory hemichannel state (for Cx 31.3) [23,31,32,42].

Connexin channels mediate the passage of substances with molecular weights not exceeding 1.2 kDa: ions (Ca^2+^, K^+^, bicarbonate), secondary messengers (inositol 3-phosphate, cAMP), reactive oxygen species (ROS), small molecules (glucose, amino acids, nucleotides, ATP, NAD^+^), peptides and microRNAs [43]. In normal tissues connexins execute tissue integration by transmitting electrical and chemical signals, maintain cell polarity, sustain differentiation, and provide nutrients and oxygen to non-vascularized tissues such as lens [9].

A plethora of experimental works on the role of connexins in cancer are being published. A lot of evidence demonstrates the tumor-suppressive role of connexins (particularly of Cx43 [44]), underpinning the seminal work of Lowenstein and Kanno on the sustained electrical integration of normal tissues [45]. However, a lot of data on the pro-tumor properties of connexins have been collected, indicating that the role of connexins in cancer is ambiguous [5,43,46]. In this review, we will discuss the role of connexins in the formation of the tumor microenvironment and, therefore, their contribution to carcinogenesis, tumor progression, and metastasis.

## 2. Connexin Localization and Its Role in Cancer

Connexin junctions directly connect cells in the TME, so connexin localization in the cell is an important issue as it determines cell behavior (Figure 2). The localization of connexins at the cell membrane can be considered in terms of their colocalization with other junctional proteins, contributing to cell polarity and cell surface protrusions. Connexins can also be localized intracellularly and be membrane-associated (in mitochondria) or represented in soluble form. A special case of localization of membrane-associated connexins is in exosomes.

### 2.1. Connexin Crosstalk with Other Adhesion Proteins

Connexin channels usually assemble at highly organized gap junction plaques and are associated with cholesterol and sphingolipids in lipid rafts [47]. Besides connexins, these plaques may include other types of junctional proteins, both integral and adapter, thus establishing a crosstalk of different types of junctions. The organization of correct adhesion complexes is mostly associated with connexin stabilization at the membrane and the formation of the less malignant phenotype of cancer cells; however, this depends on the type of the accompanying protein. For example, E-cadherin, a prominent epithelial marker, was reported to preserve the functional membrane-associated gap junctions comprising different connexins in non-transformed epithelial cells (Cx43) [48], in human squamous carcinoma (Cx43) [49], and in colorectal cancer (Cxs 26, 32, 43) [50]. N-cadherin, a mesenchymal marker, attenuates Cx43 gap junction assembly in non-transformed cells by causing its internalization via the clathrin-independent pathway [48] and at the same time, its colocalization with Cx43, necessary for successful Cx43 membrane trafficking in the advanced stages of breast cancer to maintain cancer stem cell dormancy in bone metastatic niches [51]. Earlier studies of the migration of neural crest cells revealed that loss of either Cx43 or N-cadherin impaired the directionality of cell movement [52]. Similarly, the disruption of properly arranged N-cadherin and Cx43 contacts between endotheliocytes and pericytes leads to vessel destabilization and the promotion of angiogenesis [53]. The loss of the desmosomal protein desmoglein 1 led to PKC-mediated phosphorylation of Cx43, triggering its turnover and hence a lack of cell coupling leading to skin lesions [54].

In sum, the direct interaction and mutual regulation of different junctional proteins, their contribution to cell polarity and participation in motility, when disrupted can constitute dysplasia-promoting conditions in tissues. This can lead to tissue malignization and in well-established cancer helps to adapt to variable conditions within the tumor and participate in tumor heterogeneity. A plethora of controversial evidence on the role of different types of intercellular junctions and cell–ECM contacts in cancer have conceived the view that each of them is a double-edged sword [43,55,56,57]. It seems that these contradictions can be resolved by simultaneous evaluation of the spatial, temporal and functional interplay of different junctional molecules in cancer and stromal cells, similar to what has been conducted during early embryo development studies.

### 2.2. Connexin on the Tips of Tunneling Nanotubes

Cx43 often localizes to the tip areas of thin cell surface protrusions, tunneling nanotubes (TNTs). TNTs possess an actin core, their length reaches several cell diameters, and the diameter does not exceed 1 µm. TNTs are common in aggressive tumors at advanced stages, they directly connect distant cells within the TME and thus are involved in their coordination, development of resistance of normal and malignant cells to necrosis and apoptosis, act as a means to overcome hypoxia and mediate resistance to therapeutics by rapid drug efflux [58,59,60,61]. Cx43 is widely reported to participate in the regulation of TNT formation and to be the physical connector of distant cells. In these circumstances, functional gap junctions act as cancer allies. Loss of Cx43 expression is associated with a significant decrease in the length and number of TNTs in breast cancer cell lines [62]. In line with this, secreted factors present in the conditioned medium stimulate TNT formation more effectively when derived from Cx43-expressing cells than from knockout cells. Moreover, Cx43 is involved in the crosstalk between the RhoA kinase (ROCK), protein kinase A (PKA), focal adhesion kinase (FAK), and p38 signaling pathways. It was reported that inhibition of these pathways can induce TNT formation in breast cancer cells [62]. Cx43 can also promote the formation of filopodia in HeLa cells by remodeling the actin cytoskeleton via the activation of the p21-activated protein kinase 1 and MAPK p38 pathway [63,64]. 

In MCF-7 breast cancer cells, TNTs were reported to be an important means of spheroid formation in low-adhesive conditions in vitro. At its early stages, TNTs enable physical adhesion in suspended cells, establishing Cx43 contacts with further E-cadherin expression [65]. Similarly, it has been demonstrated that Cx43 is the first echelon molecule in the self-assembly of spheroids in KGN granulosa cells, MCF-7 cells, and normal human fibroblasts [66]. 

Taking into account that Cx43 positively regulates TNT formation, one can assume that it mediates not just the opportunistic connection of two initially juxtaposed cells but also the active seeking of ‘partner cells’. Cx43, located on the tips of interpericyte TNTs, regulates the coupling of different capillary systems within the brain niche by the controlled propagation of Ca^2+^ waves [67]. Pericyte TNTs are also involved in both normal and tumor angiogenesis by the active seeking of coupling partners, be it pericytes or endotheliocytes [68]. In cancer, such active seeking is relevant in conditions of low nutrient availability and in the need for the outflow of metabolites, reminiscent of the normal developmental processes of tissues.

### 2.3. Connexin Localization in Exosomes

Exosomes are extracellular lipid bilayer vesicles with an average size of 100 nm produced by most eukaryotic cells and carry bioactive cargo of various nature (proteins, lipids, miRNA, etc.). Tumor-derived exosomes are one of the major contributors to the TME and long-distance cancer signaling [69,70]. Interestingly, exosomes can also act as chemotactic stimuli for TNT formation in cancer [71]. 

Connexins have been indicated to localize in exosomes, and associate (i.e., colocalize) with exosomal characteristic proteins, tetraspanins [72]. Exosomal connexins can facilitate the release of the exosomal contents into recipient cells and thus enhance the malignant potential, as was shown for Cx43 [73,74] and Cx46 [72]. Exosomal Cx43 facilitated colony formation and enhanced cell migration, determining the progression of glioma [74]. Analogously, it was shown that recipient cells which encountered Cx46 exosomes acquired enhanced invasive and migrative phenotypes [72]. 

Connexins may determine the mechanism of exosome loading with RNA and DNA. Using bioinformatic analysis, it has been suggested that Cx43 and Cx26 possess RNA- and DNA-binding motifs, potentially important for the transmission of genetic information between cells through extracellular vesicles [75,76]. Cx43 might participate in both the docking of exosomes with recipient cells and the transmission of the exosomal contents. 

The role of connexins in exosomal pro-tumorigenic properties is still to be elucidated, but now one can assume that it is mostly determined by the rapid cargo release into recipient cells which can either enhance the malignant potential of cancer cells or precondition a metastatic niche. Interesting speculation about the role of exosomal connexins in cancer signaling was recently suggested by Shimaoka and colleagues: Connexin channels may only release a small portion of the exosomal contents into recipient cells and this event is the decisive point in determining whether a cell will uptake the whole exosome contents by endocytosis or not [77].

### 2.4. Connexin Hemichannel Functioning

Connexin functioning as a hemichannel has a separate contribution to the TME and the development of cancer. Hemichannels can participate in the uptake of beneficial molecules (glucose, survival factors) or provide the uptake of glutamate, which promotes cell survival under nutrient deficient conditions [78]. One of the major roles of connexin hemichannels is their participation in purinergic signaling. Thus, hemichannel-dependent purinergic signaling regulates leader cell activity and the subsequent collective migration of breast cancer cells [79]. The oxidative microenvironment of bone Cx43 hemichannels in osteocytes protect breast cancer cells presumably by the relatively high release of adenosine rather than of ATP, and adenosine is reported to promote cancer growth. Additionally, functional hemichannels can release reducing factors from osteocytes into the microenvironment and lower the oxidative stress burden on cancer cells, thus promoting their growth [80].

At the same time, under cancer treatment connexin hemichannels enhanced anti-tumor efficacy due to the activation of ATP-mediated pro-apoptotic signaling in nanoparticle-treated glioblastoma [81] and PDT-treated melanoma [82]. The tumor-suppressive properties of connexin hemichannels in the absence of therapeutic treatment can be mediated by purinergic signaling too; for example, ATP release by osteocytes inhibits breast cancer cell metastasis to bone [83].

Presumably, connexin hemichannels possess tumor-promoting properties in already well-developed TMEs and participate in tumor expansion. On the contrary, their anti-tumor properties can be performed in premetastatic niches where host cells try to combat the invasion. 

### 2.5. Intracellular Localization of Connexins

Connexin channels can also be localized on intracellular compartments, namely on mitochondria. Full-fledged versions of connexins are found in the mitochondria of a wide variety of cells, including cardiomyocytes, endothelium, astrocytes [84,85,86], and malignant cells [87,88,89]. Mitochondria cannot synthesize connexins on their own so they integrate connexins into their inner membrane from the cytoplasm of the cell through the heat shock protein (HSP)90-dependent translocase of the outer membrane (TOM20) complex pathway [90]. A bold hypothesis has recently been proposed that connexins may be involved in the communication of intercellular compartments [91]. 

One of the main signs of transformed cells is changes in the metabolic and apoptotic pathways, in which mitochondria occupy a central position. It is known that tumors tend to develop conditions similar to ischemia under chronic hypoxia. A possible way of reducing mitochondrial ROS production in such conditions depends on an increase in the phosphorylation of Cx43, which accelerates its translocation into mitochondria [92]. Moreover, Cx43 can reduce the release of cytochrome *c* and the activation of caspase 3, thereby providing an anti-apoptotic effect and increasing resistance to toxins, chemotherapeutic drugs, or ionizing radiation [87]. Additionally, the anti-apoptotic role of mitochondrial Cx43 is mediated by the conserved motif in the Cx43 structure homologous with the Bcl-2 protein group which controls cytochrome *c* release [93]. Cx30 also undergoes re-localization to the mitochondria, which was observed after irradiation and was shown to induce enhanced ATP production and protect glioblastoma cells from radiation treatment [89]. 

Despite the assumptions made about an increase in cell resistance to hypoxia due to mitochondrial connexins, several works have suggested that Cx43 is involved in mitochondrial respiration, increasing oxygen consumption. The authors suggested that an increase in the expression of mitochondrial connexin shifts cellular metabolism from anaerobic glycolysis towards oxidative phosphorylation which is associated with an increased sensitivity to oxidative stress [88,94] opposing the works discussed earlier. It should be noted that mitochondrial connexins were found in embryonic stem cells and were assumed to participate in their differentiation via the generation of transient ROS waves and the shifting from glycolysis to oxidative phosphorylation [95]; therefore, the anti-tumor effect of mitochondrial connexins may be connected with improved cell differentiation.

Connexin re-localization to the intracellular space is not necessarily aimed towards mitochondria, but it is often caused by disturbances in proper connexin trafficking due to defects in the connexin itself, impairment of other participants of the trafficking processes [96,97,98], deficiency of stabilizing adapter proteins [48] or disruption of the degradation process [99]. Such intracellular localization generally leads to the enhancement of the malignant potential of the tumor due to the failure of GJ communications or acquiring atypical functions. In the case of liver cancer, the cytoplasmic accumulation of Cx32 and Cx26 enhanced the extent of mesenchymal properties of malignant liver cancer cell lines compared with the ultimate loss of connexin expression in the most aggressive cell lines [100]. The mutation of the Cx26 C-tail leads to the disruption of its appropriate trafficking due to acquiring a Golgi-retention sequence, enhancing the tumorigenicity of head and neck squamous cell carcinoma cells [96]. 

Intracellularly localized connexins perform atypical functions. The C-tail moiety of Cx43 acts as a direct promoter of the mesenchymal marker N-cadherin [98]. Ubiquitination and degradation of Cx26 and Cx43 led to the cytoplasmic accumulation of these proteins and triggered the synthesis of the stemness markers Oct4 and NANOG and the activation of the P53/MDM2 signaling pathway, which in sum led to proliferation, epithelial-to-mesenchymal transition (EMT), and the migration of pulmonary epithelial cells derived from non-small cell lung carcinoma [97]. Observing this data, it is interesting to recall that during the first cleavage divisions of a zygote connexins are already translated at the two-cell stage, but nascent proteins stay in the cytoplasm till blastomere compaction. In humans, gap junction assembly only starts in the early blastocyst stage (>32 cells). So, one can assume that cancer cells, possessing intracellularly localized connexins, resemble some features of highly pluripotent cells at the time of divergence of blastocyst lineages and right after that divergence [101]. 

It is important to note that the direct triggers of the re-localization of connexins are, to a considerable extent, caused by features of the TME, such as hypoxia [97], changes in the profile of other adhesion proteins [48] and the elevated level of their glycosylation [102]. The prevailing pro-tumorigenic outcome of the mislocalization of connexins may be explained by adaptation to the TME due to metabolic optimization, escaping the microenvironment by triggering EMT and migration, and acquiring a less-differentiated phenotype with stemness features often associated with dormancy.

## 3. Connexin Participation in the Structural and Functional Integration of Malignant, Stromal, and Immune Cells within the Tumor 

The coordinated actions of TME elements is one of the putative factors which determine tumor progression. Here, we intend to focus on the coordinative features of connexins in an awry tissue context, from the perspective of the functional and morphological aspects of the organization of the cells in the tumor tissue.

### 3.1. Connexins Integrate Tumor Cells

Connexin channels are direct bridges between neighboring cells, thus the presence of functional gap junctional plaques allows us to consider a network of such cells as a functional “syncytium”, somewhat similar to cardiac tissue [103]. The formation of such a structure provides metabolic cooperation and a platform for rapid signaling, both applicable to the cancer microenvironment to adapt to limitations. For example, such cooperation can rapidly regulate cell sizes in the actively proliferating tumor. Cells deep in the tumor experience great solid stress, thus their size is limited so water can be transported to cells of the outer layers accompanied by ion transport which proceeds through gap junctions. This causes swelling of the outer layer cells leading to increased cell proliferation [104]. Cx43-mediated glucose transfer reduces the size of the necrotic core in spheroids of colon cancer cells and elevates oxygenation and higher level of oxidative phosphorylation [105]. Cx43 located on the tips of TNTs possess integrative properties due to their capability to unite separate cells, as was indicated in vitro, such physical integration is relevant for the formation of metastatic foci and participates in angiogenesis due to the integration of endotheliocytes and pericytes (see Section 2.2).

### 3.2. Connexins Integrate into the Bone Metastaic Niche

Connexins coordinate the bone metastatic niche and lead to successful metastasis establishment due to the interaction of malignant cells with the osteoclast syncytium. In the osteoblast-conditioned microenvironment, membrane-localized Cx43 mediates tumor cell chemotaxis via its non-channel functions. At the leading edge of migrating cells, Cx43’s C-tail interacts with Rac1 and contractin, thus sustaining cell migration towards the osteogenic metastatic niche [106]. The direct interaction of cancer cells with osteoblasts through Cx43-based gap junctions provides a Ca^2+^ influx to the cancer cells enhancing their malignant potential [107]. By interacting with the osteoclast syncytium, cancer cells promote bone resorption during which a massive release of calcium and transforming growth factor-beta (TGFβ) takes place. TGFβ up-regulates Cx43, thus elevating the intracellular concentrations of calcium and enhancing GJ intercellular communications; in turn, Cx43 accelerates metastasis in the framework of this vicious cycle [108]. Taken together, the participation of connexin in bone metastasis may be considered a two-stage process with the connexin C-tail-mediated attraction of cancer cells and channel-related progression. At the same time it should be noted that in the bone microenvironment osteocytes perform purinergic signaling and can create an oxidative microenvironment by Cx43 hemichannel activity, thus combating tumor invasion [80,83]. Presumably, the overall role of connexins in bone as the potential soil for metastasis might depend on the ratio of the microenvironment participants.

### 3.3. Connexins in the Brain Niche

In the brain microenvironment, connexins appear to be the instrument that cancer cells use to turn astrocytes to allies. The connexin C-tail-mediated inhibition of Src kinase down-regulates β-catenin, triggering the preferential differentiation of neural progenitor cells towards astrocytes (against neurons). While the establishment of connexin GJs between normal neural cells and tumor cells can attenuate cell proliferation due to miR-124-3p transfer [109], astrocytes are reported to possess pro-tumorigenic activities in the glioma microenvironment [110,111,112]. One of the possible reasons for this is the transfer of miRNA derived from glioma cells to astrocytes, which promotes pro-invasive behavior, as was shown for miR-5096 [113] and miR-19b [114]. In the case of miR-19b, invasiveness might be associated with the disruption of cell–matrix adhesion. The resulting shift in the microenvironment drives tissue malignization [115]. 

Connexins heavily contribute to successful brain metastasis. Cx43 GJ communication mediates the extravasation of cancer cells to the brain parenchyma via a transcellular way. This process is initiated by the interaction of cancer cells with endothelium. The extravasated tumor cells engage Cx43 contacts with astrocytes in favor of the formation of metastasis and then form Cx43 contacts with each other, thus establishing a metastatic node [116]. The preferential interaction of tumor cells with astrocytes has been shown to be promoted by c-MYC (cellular myelocytomatosis, a proto-oncogene) [117]. Thus, in the brain niche connexins may have anti-tumor properties in the case of early tumors, while in the case of metastasis from tumors of other organs, connexins of the brain microenvironment mostly provide the means for tumor cells to pass through the blood–brain barrier and accommodate in the brain parenchyma.

### 3.4. Connexins Mediate Interactions of Cancer Cells and Cells of the Immune System

An important aspect of the integrative functions of connexins in the TME is their contribution to the communication of tumor cells and cells of the immune system. Connexins participate in immunological synapses [118] and execute both pro- and anti-tumor activities [119]. The anti-tumor activity of connexins mostly consists of their involvement in antigen presentation. Cx43 has been shown to accumulate at the interfaces between dendritic cells (DC) and cytotoxic immune cells, such as natural killers and cytotoxic T-lymphocytes, and between cytotoxic immune cells and target cells. These interactions are key to the activation and execution of the cytotoxic functions towards tumor cells [120,121,122,123]. Of note, the connexin-mediated cytotoxic effect can also be directed towards normal cells. The transfer of peptides across the gap junctions between melanoma cells and endothelial cells, for example, may provoke the destruction of endotheliocytes by cytotoxic T cells, thus hindering tumor neovascularization and lowering the supply of oxygen and nutrients to growing tumors [124]. The important anti-tumor role of Cx43 is induction of phagocytic activity of TAMs against tumor cells presumably due to the reversal of the TAM phenotype from M2 (wound-healing phenotype) to M1 (phagocytic phenotype) which was shown for melanoma [125]. At the same time connexins are one of the key players in the tumor escape from immune surveillance. Due to the transfer of signaling molecules through connexin channels, tumor cells are able to suppress the activity and cytotoxic effect of leukocytes, thus promoting the development of an immune-resistant tumor phenotype. For example, in melanoma, hypoxic tumor cells transfer miR-192-5p to DC and tumor-associated T-lymphocytes via Cx40 contacts suppressing the cytotoxic activity of T-lymphocytes [126]. In pancreatic ductal adenocarcinoma Cx31 channels mediate the recruitment of neutrophils to liver metastases and induce their polarization and survival by cAMP transfer. Neutrophils, in turn, can release immunomodulators and chemokines to enhance tumor cell colonization and immune escape [127]. 

Immune cells can also participate in the microenvironmental adaptation of the tumor. It has been suggested that tumor-associated macrophages (TAMs) may begin to act as intermediaries of nutrients between the tumor and vessels, creating an extensive cellular network integrated by Cx43 channels. The resulting rise in tumor metabolism not only increases its adaptability, but also leads to higher aggressiveness [58]. 

The connexin-mediated cooperation of stromal cells with immune cells can contribute to tumor progression and manifests as a failed regeneration process. In the case of the giant-cell tumor of bone, Cx43 participates in the fusion of osteoclasts and monocytes. This abnormal cellular cooperation is most likely associated with an attempt by monocytes to regenerate the cancer-associated damage to bone tissues, which instead leads to an increase in the size of the tumor [128].

### 3.5. Connexins in the Interactions between Tumor Cells and Other Types of Stromal Cells

Connexin contacts are established between various stromal cells, characteristic of certain metastatic niches or immune cells. Fibroblasts are a vital tissue component which substantially participates in the TME. The participation of connexins in integrating fibroblasts into the TME is tumor suppressive in the case of the early tumor and pro-tumorigenic in the advanced stage. Normal fibroblasts of the skin inhibit keratinocyte colony formation by the establishment of Cx43 GJ intercellular communications. When normal fibroblasts are exposed to sublethal doses of hydrogen peroxide (e.g., in aging) they lose Cx43 expression and thus GJ intercellular communications, which leads to the activation of keratinocyte colony formation. When the transformed fibroblasts start to prevail in the tissue microenvironment this is the turning point towards tissue malignization [129]. Interestingly, during the development of melanoma, where Cx43 is lost in malignant cells, Cx43, Cx26 and Cx30 expression was reported to be enhanced in the surrounding normal epidermis and correlated with the grade of the tumor, while this was not observed in benign nevi, as shown by measuring the mRNA content [130] and the evaluation of the protein localized at the membrane [131]. Such an up-regulation of Cx26 and Cx30 occurs during wound healing and triggers keratinocyte proliferation [130]. It should also be noted that high levels of Cx43 is characteristic for basal keratinocytes, while during maturation the level of Cx43 decreases and Cxs 26, 30 and 31 start to increase, yet their levels are still relatively low [132].

Cells of squamous cell carcinoma down-regulate homologous GJ intercellular communications mediated by Cx43 between fibroblasts in vitro in a paracrine fashion with the involvement of calcium signaling [133]. In the same type of cancer in vivo, Cx43 expression was observed in cancer-associated fibroblasts (CAF) located in the perlecan-rich stroma which is characteristic of invasion. Interestingly, these fibroblasts were localized in areas deficient in vasculature, i.e., areas of limited oxygen and nutrient supply [134]. Tumor cells can also use fibroblasts as energy donors, triggering the process of cytoplasmic and organelle sharing between fibroblasts and melanoma cells by TNT formation [135].

Initial connexin contacts between tumor cells and non-fibroblast cells can induce profibrotic properties in them, the most prominent of which is the enhanced expression of type I collagen. This was indicated for stellate cells in hepatocellular [136] and pancreatic [137] carcinomas. 

Apparently in the TME as long as connexins are predominantly used by non-tumor cells to successfully retrain tumor cells rather than by tumor cells to subjugate normal cells, they thus act as tumor suppressors. When normal stromal communication is lost, tumor cells multiply and start to prevail in the microenvironment and thus connexins become a means of tumor progression. 

## 4. Involvement of Connexins in Cancer Initiation

Early studies in connexin-deficient mice revealed their increased susceptibility to carcinogens [138,139,140,141]. This phenomenological coincidence strongly indicated that connexins are involved in cancer initiation. Indeed, Cx43-knockout mice tended to be statistically more predisposed to developing lung cancer induced by urethane or DMBA (7,12-dimethylbenz[a]anthracene) than the wild-type mice [138,139]. Cx32-deficient mice had an increased incidence of liver tumors after exposure to chemical carcinogens (DEN, diethylnitrosamine) and radiation (X-rays), with a higher number and sizes of tumor nodes compared to wild-type mice [140,141]. 

Shifts in overall connexin abundance, but predominantly its loss, is associated with tissue malignization under the action of various transforming factors, such as metabolic disorders, inflammation, bacterial infection, etc. Thus, non-alcoholic hepatosteatosis can be accompanied by the down-regulated expression of Cx32 which eventually causes liver fibrosis and is followed by hepatocellular carcinoma [142]. In the inflammatory microenvironment, for example, in the presence of pro-inflammatory cytokines (TNF-α, IL-1β and IL-6) during aging [143] or prostaglandin E2 [53], the level of Cx43 decreases. Pre-tumorigenic cells which have lost Cx43 are susceptible to the inflammatory microenvironment and they acquire a motile phenotype [144]. Improper diet and related metabolic stresses are shown to induce malignant transformation of the intestinal epithelium. In this case, activation of peroxisome proliferator-activated receptor delta (PPARD) occurs, followed by the activation of beta-catenin signaling which eventually leads to the pro-invasive elevation of Cx43 levels [145]. The well-known carcinogenic bacterium *H. pylori* can affect connexins 26, 32, 37, 43 by triggering various signaling pathways (p38 MAPK, JAK2/STAT3) or through epigenetic action on DNA (methylation of promoters, acetylation of histones), thus regulating the expression levels of these molecules, governing their localization or creating their polymorphism [146,147]. 

The improper membrane localization of connexins can also become a tumor-initiating event. Infection with human papillomavirus 16 causes mutation in the human homolog of Drosophila discs large (hDlg)-binding motif of the Cx43 C-tail, which results in the disruption of proper Cx43 and hDlg trafficking to the membrane so Cx43 remains in the cytoplasm [148]. 

Another mechanism is the initiation of endometrial cancer in obese patients which can be due to the disruption of Cx43 GJ intercellular communications by hypermethylation of the Cx43 promoter in normal endometrial epithelium mediated by the microenvironment conditioned by adipose stromal cells which massively release plasminogen activator inhibitor 1 [149].

Considering not only the levels of individual connexins, but also taking into account their interplay with each other, relative changes in their levels are also important. For example, the turning point for the transformation of bile duct epithelium by *Clonorchis sinensis* metabolites and eventual cancer development is accompanied by the simultaneous decrease in the expression of Cx32 and up-regulation of Cx43 and Cx26. Cx43 in this case acts as a tumor promoter as its inhibition leads to a proliferation decrease [150]. Similarly, the altered expression and functionality of Cx43 and Cx32 in liver oval cells have been demonstrated to cause hepatocellular and cholangiocellular neoplasms due to the disruption of adequate cell differentiation [151]. This evidence may be supported by the fact that differentiation towards normal liver tissues in early development requires the sustained expression of Cx32 and down-regulation of Cx43 [152,153].

Another way to alter intercellular communication and hence the tissue context is the stable expression of connexins which are normally only transiently expressed in a tissue when necessary for precise functional needs. For example, in normal breast development Cx32 is expressed only during lactation precisely at the interfaces of luminal cells [154] but this is characteristic of metastatic breast cancer lesions in lymph nodes. Overexpression of Cx32 in the normal breast epithelium cell line MCF10A turned the cell morphology towards a mesenchymal phenotype and increased the migratory activity of these cells by triggering the expression of EMT markers [155].

A rare example of non-channel connexin-dependent carcinogenesis is supposed to be associated with de novo expression of connexins, initially foreign to the tissue. For example, expression of Cx46 is not detected in normal breast epithelium and is detected in early breast cancer. De novo expression of Cx46 in breast epithelium was phenomenologically demonstrated to protect cells from hypoxia using xenograft models in vivo. The authors assumed that it may be a cancer-initiating event [156].

Summarizing the represented data we may consider that the tumor-initiating factor, or, to be more precise, a complex of conditions in the site of tumor initiation, forces perturbations in the functioning of connexins, namely, decreasing in their amount, re-localization, shifting the ratio of connexins typical to the tissue, establishing stable expression of certain connexins which are transiently expressed in the tissue, or the de novo expression of connexins which are normally foreign to it. It is necessary to admit that these data are mainly phenomenological, and a mechanistic explanation of how the switch in the connexin profile promotes malignant transformation is a matter for future research. Still some scattered pieces of this picture can be seen already. The loss of connexins characteristic to the tissue triggers the up-regulation of oncogenes, such as c-myc [157,158,159], cyclin D1 [157], and stemness markers, such as CD133, Nanog and Oct4 [158,159]; and the overexpression of non-typical connexins also leads to the up-regulation of oncogenes, such as EGFR [160] and CTNNB1 [161]. Possible answers may be found in comparing the intercellular communications in a tumor with different stages of a given tissue’s early development.

## 5. Role of Connexins in Forming a Hypoxia-Resistant Cancer Cell Phenotype

Hypoxia is a biological condition characterized by insufficient oxygenation of tissues which, in the case of malignant tumors, is caused by rapid cell proliferation [162,163]. The oxygenation levels in tumors are lower compared to normal tissues. For example, it was demonstrated with the polarography method, that the oxygenation levels in well-oxygenated tissues, such as muscle, lie in the range of 20 to 70 mmHg, while in breast tumors it ranges between 3 and 30 mmHg, which indicates hypoxia [164]. 

Hypoxic conditions are, on the one hand, a circumstance that is destructive to tumor cells, which cells must cope with, and on the other hand, a powerful selection factor for their collective adaptation with the subsequent development of progression mechanisms. Connexins participate in both of these aspects. 

Overcoming hypoxia on a large scale is realized in tumors by promoting angiogenesis mainly by establishing the proliferative and migrative phenotype of endothelial cells. This can be realized via different types of cell–cell interactions, i.e., between tumor cells, tumor cells and endotheliocytes and between stromal cells, including vascular cells, and is mostly provided by functional GJ intercellular communications. For example, transcriptional suppression of Cx43 and Cx26 in MDA-MB-231 breast cancer cells led to the down-regulation of GJ intercellular communications between cancer cells and cancer cells-and-endotheliocytes, accompanied by reduced migrative and invasive properties, as shown by real-time cell analysis [165]. These results suggest that GJ intercellular communications between tumor cells and endotheliocytes enhance their migration and proliferation. It has been shown that collectively migrating tumor cells that have formed Cx43 contacts with endothelial cells, i.e., pre-hypoxic micrometastases, trigger vascularization upon the onset of hypoxic conditions [166]. On the other hand, GJ intercellular communications between endothelial cells is vital for initial vessel integrity. Loss of Cx43 expression and hence its patchy membrane localization leads to increased permeability of the existing vessels and promotes angiogenesis, as shown for high-grade serous ovarian cancer [167]. Vascular permeability is reported to be the factor stimulating angiogenesis [168]. Another mechanism of angiogenesis stimulation due to Cx43 loss relies on the massive production of pro-angiogenic factors due to the elevated levels of hypoxia-induced factor alpha 1 (HIF1a), as Cx43 is responsible for HIF1a ubiquitination and degradation indicated in melanoma [169]. 

One of the characteristics of a severely hypoxic microenvironment that should be handled is acidosis, and functional GJ intercellular communications mediated by connexins allows its management. Thus, in spheroids of pancreatic cancer, it has been shown that connexin channels between hypoxic and normoxic cells allow the rapid distribution of bicarbonate ions to neutralize acidification in hypoxic areas [170]. More than this, tumors can protect themselves from acidification by another mechanism, specifically, transmitting lactate through Cx43-based channels [171]. It has been shown that acidosis management can be also carried out with the participation of the stroma. Hydrogen ions, produced by tumor cells, are captured from the extracellular space by the AE2 transporter on myofibroblasts and are then spread via Cx43 channels through the myofibroblast syncytium [172]. The transfer of ions occurs passively, which is energetically beneficial for hypoxic cells, as they retain ATP; moreover, a spread of lactate can additionally act as an alternative nutrient shared between cancer cells, as lactate is indirectly involved in the tri-carbon acid cycle [173]. Although as demonstrated in severe acidosis, when lactate concentrations are too high and high calcium concentrations are established, Cx43 channels close and uncouple, which leads to the interruption of the GJ intercellular communications in rat hepatocellular carcinoma and human glioblastoma A172 [174].

Thus, connexins participate in hypoxia resistance by providing a trans-cellular path for oxygen and nutrients in arranged cells which jut into the depths of the tissue from the perivascular space [58] or by providing the spread of alternative nutrients (lactate) simultaneously protecting cells from acidification. The loss of GJ intercellular communications in vascular cells triggers angiogenesis; and the re-establishment of GJ intercellular communications in impaired vessels facilitates the migration of tumor cells through the vessel wall, enabling the formation of metastatic units already possessing resistance to hypoxia.

## 6. The Multifaceted Role of Connexins in Both Tumor Progression and Suppression Due to the Intracellular Transfer of miRNA

miRNAs are one of the most potent factors of the TME and they are transmitted between both malignant and stromal cells through connexin-based gap junctions or via extracellular vesicles (exosomes). Thus, Cx43-based channels have been reported to transmit miRNA-145 from microvascular endothelial cells to colon cancer cells leading to the inhibition of angiogenesis [175]. A similar cancer-inhibiting effect was recently reported in glioma where the miR-152-3p transmitted from normal astrocytes to C6 glioma cells via Cx43-based channels attenuated their migration and invasion [176]; and in hepatocellular carcinoma where miR-142 and miR-233 were transferred from macrophages to tumor cells [177]. In glioma cells loaded with tumor-suppressive miR-124-3p, GJ intercellular communications enhanced the transfer and distribution of miRNA to neighboring cells which attenuated cell proliferation, as was shown in vitro and in vivo [109]. 

A considerable amount of data has been gathered on the pro-tumorigenic role of miRNA transfer by connexin channels. The miR-5096 derived from glioma cells possesses a pro-invasive effect when transferred to astrocytes [113], and a pro-angiogenic effect when transferred to microvascular endothelial cells along with the suppression of Cx43 expression [178]. Hypoxia-induced miR-192-5p transferred through Cx43-channels from melanoma cells to cytotoxic T-lymphocytes triggers the immune surveillance escape [126]. Bone marrow-derived miRNAs targeting CXCL12 were indicted to contribute to breast cancer quiescence [179].

In the case of the exosome-mediated transmission, connexin channels have been reported to recruit miRNAs to exosomes as they possess RNA-binding motifs in their structure [75]. The miRNA transmission mediated by exosomes containing connexins facilitates cancer progression in hypoxic conditions (Cx46-rich exosomes) [72]. It is interesting to note that Cx43-based channels have a higher permeability for various miRNAs compared to channels formed by other connexins, such as Cx26, Cx30, and Cx31 [180], or Cx32 and Cx37 [181]. 

Summarizing these data, the trend towards tumor progression or suppression may be determined by the type of miRNA which is being transferred and the direction of transfer (from a tumor cell to a normal cell or vice versa, or between tumor cells with different malignant potential). This can probably be determined by the presence, quantity, affinity, and conformational availability of RNA-binding motifs in various connexins.

## 7. Connexins and Cancer Stemness

Cancer stemness is an important factor of cancer progression, which determines cancer self-renewal, dormancy, and resistance to treatment [182]. Connexins are reported to participate in regulating cancer stemness in positive and negative ways, or utilized by cancer stem cells to perform their functions.

Connexins can reduce the stemness of cancer cells or provide assistance in cancer treatment aimed at resistant stem cells. Thus, ectopic expression of Cx43 in lung cancer cells reduces the abundance of cancer stem cells, as was shown by a reduction in tumor sphere formation and stemness markers in transfected cells [183]. Stemness attenuation was also reported for Cx30 in glioma due to its ability to interfere with the insulin-like growth factor 1 receptor, which is involved in maintaining self-renewal [158]. The assistance in coping with cancer stem cell drug resistance consists of the establishment of GJ communications, which was indicated in liver cancer, where simultaneous ectopic expression of Cx43 and SUMO1 resulted in a higher responsivity to treatment [184].

Connexins can also support the stemness features of cancer cells. The most prominent stemness feature supported by them is self-renewal which was established in the case of intracellular localization of connexins. For example, it has been shown that Cx26, Cx32 and Cx46 can form alternative signaling with the pluripotency transcription factor NANOG, up-regulate CD133 or markers of stemness Oct4 and Sox2, which leads to an increased cancer abundance of stem cells and the acquisition of an invasive tumor phenotype [185]. The functional GJ communications can also support stemness features, as was indicated for Cx46, which maintains self-renewal in glioblastoma [186]; and for Cx43 which is crucial for maintaining pluripotency and proliferation in embryonic stem cells [187] and maintaining dormancy in the bone marrow niche [51]. 

Connexins can also be used by cancer stem cells as a means of the realization of their aggressiveness. Thus, Cx43 was reported to mediate breast cancer immune escape by establishing communications between cancer stem cells and mesenchymal stem cells which results in a preferential Treg response against T-helper 17 cells [188]. Additionally, Cx43 facilitates lung cancer metastasis to the brain by establishing communications between cancer stem cells and astrocytes [189].

The recent conceptual papers by J.E. Trosko discussed the hypothesis that connexins may act as key molecules which underlie cancer stem cell origin and determine the cancer stem cell type due to their tissue integration properties with the most crucial point, cell differentiation. Cancer cell stemness, in this case, is assumed to be maintained on the one hand by the sustained expression of the Oct4A oncogene which prevents the expression of connexin genes, thus making it impossible for cells to differentiate at all, or in the case of when Oct4A oncogene is absent the stemness is maintained by other oncogenes which prevent proper connexin localization and the execution of their differentiation functions [190,191,192].

## 8. Conclusions

The holistic picture of cancer origin and development is still an unsolved issue. Accumulating data indicate that tissue context is crucial in this process. In the 1990s a tissue-organization field theory (TOFT) of cancer origin and development was suggested, which postulated that cancer is a tissue disease and that it is based on the disorder of cell organization, rather than one renegade cell [2]. So, based on this theory, it can be assumed that intercellular interactions can be one of the key factors that determine tissue integrity in normalcy, and, if their work is disturbed, they can have a significant contribution to tissue disorganization. Connexin gap junctions integrate tissues, maintain their polarity, and the connexin proteins also possess non-channel-related tumor-suppressive properties. In normal epithelial tissues their abundance and distribution are very precise (Figure 3A), and match with the level of cell differentiation (e.g., in skin epithelium) [132], or is modulated during the functioning of secretory epithelium (e.g., in mammary gland) [193].

Loss of functioning connexin channels, or, as a worse outcome, mislocalization of connexin can lower the level of tissue orderliness. The disbalance in the abundance of connexins which are characteristic of the tissue, or stable expression of connexins which are only needed for precise functional states of the tissue (e.g., Cx32 in mammary epithelium [154]), or even the de novo expression of connexins which are not normally detected in that type of tissue (e.g., Cx46 in mammary epithelium [156]) can act as a cancer-initiation factor (Figure 3B). This can be, on the one hand, due to the loss/disturbance of integrative functions executed by GJICs, and on the other hand due to the acquisition of a motile phenotype by such cells due to the expression of EMT markers. 

Connexins are indeed found to be involved in regulation of EMT. For example, Cx43 can reverse EMT, as its overexpression leads to the elevation of epithelial markers, E-cadherin and ZO-1, and the down-regulation of mesenchymal markers, vimentin, Snail and Slug [194,195]. This occurs due to the inhibition of MAPK/ERK signaling [196] and Akt signaling [195]. In hepatocellular carcinoma down-regulation of Cx32 led to an increase in vimentin and Snail and a down-regulation of E-cadherin [197]. Interestingly, overexpression of Cx32 in breast cancer cells with Cx32 intracellular localization led to the slight reversal of EMT (E-cadherin and ZO-1 levels were elevated, and N-cadherin was lowered), but the Snail expression was increased and proliferation enhanced [198]. In breast tumors the expression of Cx46 (atypical connexin for breast tissue) increased the expression of EMT markers, namely, N-cadherin, vimentin, Snail and Zeb1, and stemness markers [199]. It should be noted that the expression of EMT markers elicited by perturbations in the profile of connexins may not be a sign of the transition of a cell to a completely mesenchymal state, because the profile of other adhesion proteins in the cell and EMT-related proteins can also vary. This demonstrates that these perturbations may contribute to the so-called partial EMT, when these cells possess both epithelial and mesenchymal traits. This provides greater cell plasticity compared to full EMT, and it relies on the internalization and re-localization of adhesion proteins [200].

The interactions mediated by connexins between tumor cells and anti-tumor stroma result in the restriction of tumor growth and is most clearly represented by immunological synapses [120,121], interactions with normal tissue fibroblasts [129], and normal astrocytes [176] (Figure 3C). Still, the anti-tumor capacity of the stroma is not infinite and the persistent proliferation and migration of tumor cells can shift the precarious balance. The possible reason for turning connexins into cancer allies may arise from the establishment of heterogeneity in both tumor and stromal cells. In this case, local pro-tumor stroma will prevail thus giving the tumor a way to further progress. Connexins in this case can act as modulators of the phenotype of individual cells (by constituting EMT or stemness), or act as a means of communication in favor of the tumor (by various means, including providing a nutrient and oxygen supply from the perivascular area, by distributing acidifying ions, maintaining an immunosuppressive microenvironment, etc.).

During metastasis, connexins participate in all stages, from preconditioning the metastatic niche, establishing cell motility, enabling collective migration of tumor cells coupled with immune cells and endotheliocytes, diapedesis of the metastatic unit and homing to the metastatic site (Figure 3D).

In order to resolve the controversy of the role of connexins in cancer and precisely in the TME, a complex approach should be executed. If we assume that intercellular junctions are one of the putative means defining a multicellular organism, we should study the dynamics of the overall junctional complex of the cells in the TME, as during tissue development the role of each type of these junctions can hardly be separated. Such attempts have been made [201,202]. It should also be pointed out that this kind of research is only appropriate in a 3D microenvironment, since the abundance of junctional molecules, and connexins in particular, in monolayers is seriously altered compared to in vivo [105,201,203]. Even if the connexins abundance is equal in 2D and 3D conditions, the overall cell phenotype may be drastically different due to differences in the abundance of other junction proteins [204], or the molecular mass of connexins may be lower in vitro due to the lack of post-translational modifications which take place in vivo [205]. 

The importance of the TME in tumor progression and response to treatment has led to special attention being paid to (i) tissue-relevant tumor models used in drug discovery, (ii) clinical diagnostic methods allowing the assessment of the tissue organization of tumors in addition to its size and localization, and (iii) TME-targeting tumor treatment approaches. Thus, 3D in vitro models are increasingly favored over conventional monolayers [206,207,208,209,210]. Despite the significant progress, to date, there are no ideal models capable to completely capture the actual tumor microenvironment. Differences in experimental conditions result in controversial data being obtained for cells of identical origin [203,211], thus the search and investigation for new models is a highly relevant task. 

The list of diagnostic imaging methods used by researchers and clinicians has recently been expanded with magnetic resonance imaging (MRI)-based techniques (functional MRI, MRI spectroscopy, high-resolution MRI, etc.) providing information on the spatiotemporal pattern of tissue vascularity and perfusion, cellularity, extracellular pH within the tumor niche and other metabolic features [212,213,214]. The activity of enzymes playing a role in TME organization, for example, matrix metalloproteinases and cysteine cathepsins can be visualized with radioisotope-labeled tracers by positron emission tomography (PET) and single-photon emission tomography (SPECT) [215,216,217]. The multiple options include multiple imaging techniques to study tumor-associated inflammation and immune cell populations, hypoxia, glycolysis, matrix proteins, etc. (see for review [218,219,220].

In addition to TME assessment by imaging methods, the detailed information on the TME–connexins interplay can be obtained from cellular level analyses using sequencing and omic techniques. These approaches are becoming important instruments in TME research and clinical practice due to a growing interest in personalized medicine. Single-cell sequencing is able to delineate the heterogeneity of tumor and tumor-associated cell sub-populations, such as cancer-associated fibroblasts, responsible for the development of chemoresistance via Cx43 up-regulation [221,222], or macrophages participating in response to immunotherapy treatment [223]. The next level approach of single-cell multi-omic analysis provides the possibility to decipher the interactions among the genome, epigenome and transcriptome in tumor single cells and reveal their reaction to changes in the TME [224,225]. 

The TME and, specifically, connexins can be considered as a target for tumor treatment. To date several approaches utilizing connexins as potential therapeutic targets are being developed. Firstly, the restoration of GJICs either by combating with conditions which block it (by inhibiting the enzymes which produce blocking substances) or the restoration of the expression of connexins [226]. In metastatic tumors, where connexins act as cancer allies, GJIC inhibitors have been used. The more precise manipulation with the expression of connexins can be achieved by using connexin mimetic peptides and connexin-specific antibodies [227]. Connexins have also been studied as potential biomarkers, e.g., for survival prediction [228], tumor grading [229] and subtyping [230], or as predictive biomarkers of cancer occurrence [231]. To summarize, rapidly developing research methods and a plethora of elaborated tumor models along with state-of-the-art techniques for clinical diagnosis and data analysis may one day be translated into a set of comprehensive therapeutic strategies.

## Figures and Tables

**Figure 1 biology-12-00204-f001:**
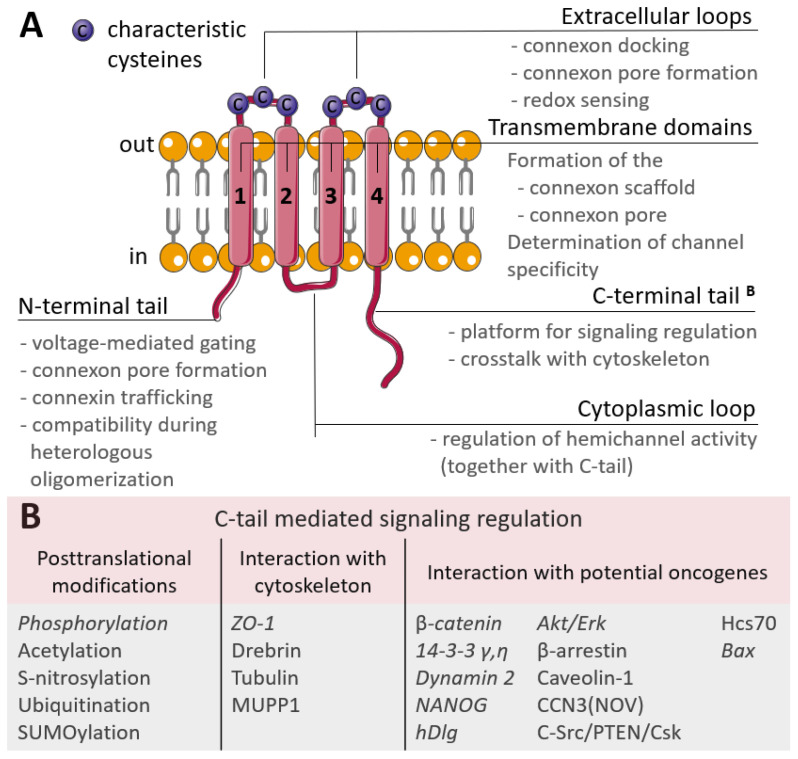
(**A**) Scheme of the connexin structure. Connexin is a tetraspan molecule which contains four transmembrane α-helices (1-4), two extracellular loops and one cytoplasmic loop; the amino- and carboxy-termini are located inside the cell. Transmembrane domains participate in the formation of the connexon scaffold and pore formation; extracellular loops are responsible for channel docking; the cytoplasmic loop and the N- and C-tails are the platform for the regulation of connexin functioning. (**B**) Participation of the connexin C-tail in regulation. The reported cancer-related signaling is represented in *italic* (explanations are in the text).

**Figure 2 biology-12-00204-f002:**
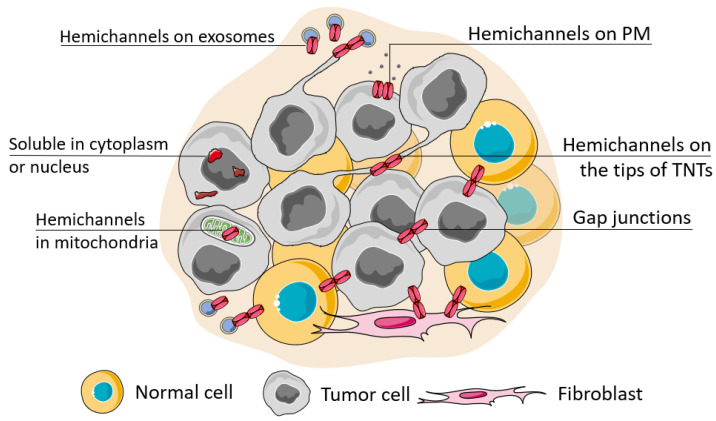
Possible localization of connexins in the cell. The localization of connexins in the cell can be attributed to its soluble forms (located in the nucleus or cytoplasm) or membrane-bound forms which can be observed during its trafficking, ultimately represented as a functional connexin residing in the cell membrane in the hemichannel or channel state (when docked with a connexin of a neighboring cell). The localization of connexin channels at the membrane can be considered relatively to their localization to the basal membrane (cell polarity), attributed to cell protrusions (e.g., tunneling nanotubes, TNTs) or extracellular vesicles (e.g., exosomes). Connexins can be also transferred to the inner membrane of mitochondria.

**Figure 3 biology-12-00204-f003:**
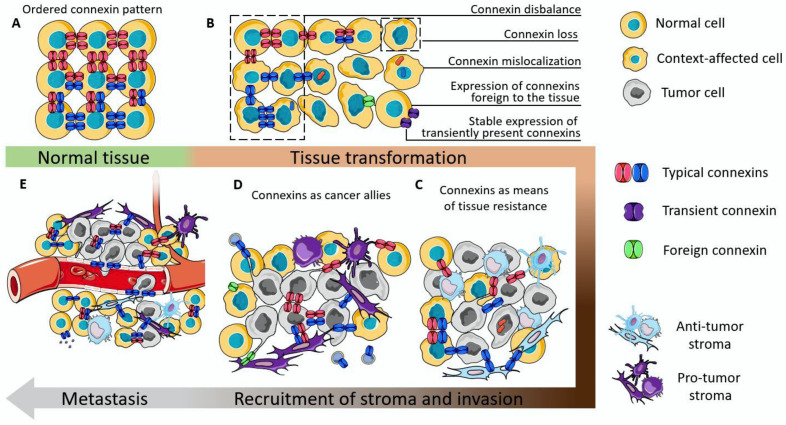
The proposed scheme of connexin participation in the initiation and development of cancer. (**A**) The schematic representation of the ordered connexin pattern in normal tissue. (**B**) Examples of connexin dysregulation which can contribute to the disruption of tissue orderliness. (**C**) Connexins as integrators of tumor parenchyma with the stromal cells which suppress tumor aggressiveness. (**D**) Connexins as integrators of tumor tissue, contributing to its progression. (**E**) Connexins as a means to perform and facilitate metastasis.

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
