# Peer review of "The Multifaceted Role of Connexins in Tumor Microenvironment Initiation and Maintenance"

_biology, 2023, doi:10.3390/biology12020204_

Round 1

Reviewer 1 Report

The scope of this article continues to be too broad (everything about connexins and everything about connexins/gap junctions and cancer) to make it a useful addition to the literature. The authors have attempted too large a scope. Therefore, there is little in this "review" that critically appraises the published literature. It is hard for me to draw any useful conclusions regarding the important roles of connexins in the tumor microenvironment.

Minor comment:

Why do the authors persist in using unnecessary abbreviations? I understand that the other reviewer corrected the connexin abbreviation to Cx. But why ever abbreviate Cx (rather than spelling out connexin) other than when there is a number attached to identify a specific isoform?

Author Response

Response to the Reviewers comments

Dear Reviewer,

We would like to express our sincere appreciation for the thorough assessment of our work. Please find below the detailed description of the revisions performed according to your remarks. (Reviewer’s comments are presented in bold, our answers follow in plain text).

The scope of this article continues to be too broad (everything about connexins and everything about connexins/gap junctions and cancer) to make it a useful addition to the literature. The authors have attempted too large a scope. Therefore, there is little in this "review" that critically appraises the published literature. It is hard for me to draw any useful conclusions regarding the important roles of connexins in the tumor microenvironment.

We thank the Reviewer for the thorough assessment of our work. In our opinion, the review article can be of two types: the “intensive” critical analysis of the state-of-the-art in the defined scientific field, and “extensive” description of the relatively large-scope area. We absolutely agree with the Reviewer that the “intensive” critical review is most useful and valuable for specialists working in the field. However, the “extensive” reviews have their own specific target audience. Such reviews are of undoubted usefulness for young researchers and Ph.D. students, members of interdisciplinary teams, universities’ staff, and scientists coming to a new scientific field following the research trajectory. Extensive reviews help this type of audience to get an idea of the overall situation, and to reveal the hot topics and crucial issues in the research area. Usually, the “extensive” review refers the reader to “intensive” works for detailed information.

We aim at such a large-scope review of the connexins' role in TME and believe that it could appeal to a wide audience.

Minor comment:

Why do the authors persist in using unnecessary abbreviations? I understand that the other reviewer corrected the connexin abbreviation to Cx. But why ever abbreviate Cx (rather than spelling out connexin) other than when there is a number attached to identify a specific isoform?

We have corrected the use of Cx abbreviation. The abbreviation is now used only when the specific isoform is mentioned, (e.g. Cx43, Cx32). In all other cases, we use the whole word “connexin”.

Reviewer 2 Report

The review paper summarizes data on the contribution of connexins (Cxs) to the formation of tumour microenvironment (TME) along with the initiation and progression of cancer.

Authors conclude to the dysregulation of the ordered Cx pattern by way of of interacting junctional proteins as a constitutent of dysplasia-promoting conditions in the tissue. They propose a concise scheme of participation of Cxs in the initiation and development of cancer. The unifying scheme starts from the ordered Cx pattern in the normal tissue through Cx dysregulation which can contribute to disrupt tissue orderliness right up to Cxs functioning as integrators of tumor tissue formation, progression and metastasis. Cxs can also function as integrators of tumor parenchyma with the stromal cells which suppress tumor aggressiveness. The reported role for Cx43 in tumor tissue genesis via the formation of tunneling nanotubes (TNT) highlights the importance of the initiation of novel drug design, discovery and development campaigns based on specific inhibition of intercellular coupling inhibitors.

Author Response

Response to Reviewers' Comments

Dear Reviewer,

We would like to sincerely thank you for such a high appreciation of our work.

Reviewer 3 Report

Pro- or anti-tumorigenic properties of connexins are determined by their abundance, localization, and functionality as well as channel assembly and non-channel functions. In this review, we have summarized the data on the Cxs contribution to the formation of TME and to the cancer initiation and progression. The following questions need to be determined.

1. What's the interaction of potential oncogenes, for example c-Myc. Does c-Myc plays an important role of it?

2. Does interleukin also plays an important role of it?

3. Please also focus on microenvironment of tumor connection. for example EMT and MET mechanism.

Author Response

Response to the Reviewers comments

Dear Reviewer,

We would like to express our sincere appreciation for the thorough assessment of our work. Please find below the detailed description of the revisions performed according to your remarks. (Reviewer’s comments are presented in bold, our answers follow in plain text).

Pro- or anti-tumorigenic properties of connexins are determined by their abundance, localization, and functionality as well as channel assembly and non-channel functions. In this review, we have summarized the data on the Cxs contribution to the formation of TME and to the cancer initiation and progression. The following questions need to be determined.

  1. What's the interaction of potential oncogenes, for example c-Myc. Does c-Myc plays an important role of it?

The loss of connexins characteristic to the tissue as well as overexpression of non-typical connexins can trigger the upregulation of several oncogenes in cells. We have added information on the interaction of connexins with oncogenes to the manuscript. Please see lines 567-571 highlighted with green color.

  1. Does interleukin also plays an important role of it?

The pro-inflammatory cytokines including interleukins can affect the level of connexins. The experimental evidence is obtained for connexin 43. We have added information on the influence of interleukins on connexins to the manuscript. Please see lines 511-515 highlighted with green color.

  1. Please also focus on microenvironment of tumor connection. for example, EMT and MET mechanism.

We have expanded the discussion on the involvement of connexins and EMT in the tumor microenvironment. Please see lines 733-750 highlighted with green color.

Reviewer 4 Report

The review represents a timely and comprehensive overview of the role of connexins in cancer with a specific focus on tumor microenvironment. The article adequately covers the biology of proteins, their biochemical structure and function and their specific activity in tumor cells and their microenvironment and how this contributes to tumor initiation and progression. It is well written and incorporates findings from several new studies. References have been correctly cited and the review would appeal to a wide-ranging audience in the scientific community.

This article would be further improved by elaborating in the importance of connexins and their associated signaling as biomarkers and therapy targets as this manifested by previously developed clinical associations. This could also be supported by the description of the signaling that regulates their expression and activity in normal and cancer tissues. Although the manuscript lists scattered information about the physiological role of the proteins, it would help the reader to see a concise description of their role in normal tissues at the beginning before moving to cancer. It would also be interesting to hear a discussion of how this signaling could be manipulated to restore cellular phenotypes and their environment as an approach to improve therapies. Although the title indicates the focus of the review on the role of connexins in cancer progression, it seems biased to not mention at all that numerous studies have also shown connexin43 can be a tumor suppressor too as detailed in Aasen et al., (2016). Nature Reviews Cancer, 16(12), 775-788. Finally, adding the number to how many different connexins have been discovered so far and which of those have been shown to have a role in tumor progression and tumor micro-environment would help the readers put things into perspective.

Author Response

Response to the Reviewers comments

Dear Reviewer,

We would like to express our sincere appreciation for the thorough assessment and such high appreciation for our work. Please find below the detailed description of the revisions performed according to your remarks. (Reviewer’s comments are presented in bold, our answers follow in plain text).

The review represents a timely and comprehensive overview of the role of connexins in cancer with a specific focus on tumor microenvironment. The article adequately covers the biology of proteins, their biochemical structure and function and their specific activity in tumor cells and their microenvironment and how this contributes to tumor initiation and progression. It is well written and incorporates findings from several new studies. References have been correctly cited and the review would appeal to a wide-ranging audience in the scientific community.

This article would be further improved by elaborating in the importance of connexins and their associated signaling as biomarkers and therapy targets as this manifested by previously developed clinical associations.

The clinical application of connexin-based therapy and diagnosis is a promising and rapidly developing area. Due to limited size of the review article we weren’t able to include the broad description of the topic. However, we emphasized the importance of connexins as the therapeutic targets and biomarkers and refer the readers to the recent reviews. Please see lines 788-796 highlighted with green color.

This could also be supported by the description of the signaling that regulates their expression and activity in normal and cancer tissues.

We have added information on the regulation of the expression of connexins, please see lines 57-61 highlighted with green color.

Although the manuscript lists scattered information about the physiological role of the proteins, it would help the reader to see a concise description of their role in normal tissues at the beginning before moving to cancer.

We have added a list of functions of connexins in normal tissues prior description of the cancer features. Please see lines 135-138 highlighted with green color.

It would also be interesting to hear a discussion of how this signaling could be manipulated to restore cellular phenotypes and their environment as an approach to improve therapies.

We have added information on the notion that connexins are elaborated as therapeutic targets and biomarkers. Please see lines 788-796 highlighted with green color.

Although the title indicates the focus of the review on the role of connexins in cancer progression, it seems biased to not mention at all that numerous studies have also shown connexin43 can be a tumor suppressor too as detailed in Aasen et al., (2016). Nature Reviews Cancer, 16(12), 775-788.

In the review, we support the idea that connexins play a controversial role in tumor containment or promotion depending on a plethora of factors. To avoid misunderstanding, we have emphasized the tumor-suppressing role of Cx43. Please see line 140 highlighted with green color.

Finally, adding the number to how many different connexins have been discovered so far and which of those have been shown to have a role in tumor progression and tumor micro-environment would help the readers put things into perspective.

We have added and emphasized this information. Please see line 46 and lines 51-52 highlighted with green color.

Round 2

Reviewer 1 Report

There have been no substantive changes in this manuscript since my prior review. I continue to find it superficial and non-critical

Author Response

We respect the Reviewer's opinion and thank him/her for the work. We believe that the comments of the Reviewer helped us to improve the manuscript.